# Incidence and Clinical Features of Pseudoprogression in Brain Metastases After Immune-Checkpoint Inhibitor Therapy: A Retrospective Study

**DOI:** 10.3390/cancers17152425

**Published:** 2025-07-22

**Authors:** Chris W. Govaerts, Miranda C. A. Kramer, Ingeborg Bosma, Frank A. E. Kruyt, Frederike Bensch, J. Marc C. van Dijk, Mathilde Jalving, Anouk van der Hoorn

**Affiliations:** 1Department of Medical Oncology, University Medical Center Groningen, University of Groningen, 9713 GZ Groningen, The Netherlands; f.a.e.kruyt@umcg.nl (F.A.E.K.); m.jalving@umcg.nl (M.J.); 2Department of Radiology, Medical Imaging Center, University Medical Center Groningen, University of Groningen, 9713 GZ Groningen, The Netherlands; a.van.der.hoorn@umcg.nl; 3Department of Radiation Oncology, University Medical Center Groningen, University of Groningen, 9713 GZ Groningen, The Netherlands; m.c.a.kramer@umcg.nl; 4Department of Neurology, University Medical Center Groningen, University of Groningen, 9713 GZ Groningen, The Netherlands; i.bosma01@umcg.nl; 5Department of Pulmonary Diseases and Tuberculosis, University Medical Center Groningen, University of Groningen, 9713 GZ Groningen, The Netherlands; f.bensch@umcg.nl; 6Department of Neurosurgery, University Medical Center Groningen, University of Groningen, 9713 GZ Groningen, The Netherlands; j.m.c.van.dijk@umcg.nl

**Keywords:** brain metastasis, immune-checkpoint inhibitor, pseudoprogression, MRI

## Abstract

There are currently no studies reporting on the incidence of pseudoprogression in brain metastases at the point of initial change in contrast enhancement suspected for tumour progression after the start of immune-checkpoint inhibition. This study addresses this question and also investigates the associated clinical features of pseudoprogression of brain metastases in this context. This study reports that the lesion-specific incidence rate of pseudoprogression is at least 9.4% in brain metastases after immune-checkpoint inhibition. The sole clinical feature distinguishing pseudoprogression from true tumour progression was that patients with tumour progression were more likely to need dexamethasone for neurological symptoms. These findings give an initial suggestion to clinicians concerning the likelihood of pseudoprogression when confronted with imaging changes suspicious for intracranial progression after initiating immune-checkpoint inhibition in melanoma and non-small cell lung cancer.

## 1. Introduction

Brain metastases occur in up to 20–30% of cancer patients [1,2,3]. Median overall survival after diagnosis is approximately 6–10 months, but this can vary greatly between patients and cancer types [3,4]. Treatment consists of resection and radiotherapy, which is generally given as stereotactic radiotherapy (SRT) [5]. These local treatments are often combined with systemic therapy, which includes targeted therapy and treatment with immune-checkpoint inhibitors (ICIs) [5]. ICI therapy has become standard of care for non-small cell lung cancer (NSCLC) and melanoma, the two tumour types most likely to lead to intracranial metastatic disease. The widespread use of ICI therapy for brain metastases from these cancer types is based on encouraging reports of intracranial efficacy in recent clinical trials [3,6,7,8]. In melanoma, complete and partial response rates of 33% and 21%, respectively, are seen with combination nivolumab/ipilimumab [6,7]. In NSCLC, around 30% of patients with programmed death-ligand 1 positive disease will have an intracranial response with pembrolizumab treatment [8].

A well-known dilemma during follow-up after SRT are changes in contrast enhancement within the irradiated region on MRI. This may represent true tumour progression (TP),but may also be a mimic due to benign inflammatory processes, blood–brain-barrier disruption and/or vascular leakage [9]. The latter phenomenon, termed ‘pseudoprogression’ (PsP), can complicate clinical decision-making [9,10,11]. Uncertainty about imaging changes can lead either to inappropriate changes in therapy or, conversely, delays in starting new treatment. PsP is also seen after ICI therapy in brain metastases, both with and without concomitant radiotherapy. It has been described in multiple case reports, usually presenting as an increase in central enhancement and perilesional oedema with associated FLAIR abnormalities [12,13,14,15,16]. Guidelines for the follow-up and identification of ICI therapy-induced PsP in brain tumours have been published by the response assessment in neuro-oncology (RANO) working group as part of the iRANO criteria [17].

Although PsP is recognised to occur after ICI therapy, its dynamics remain poorly understood. Notably unclear is its incidence and to what degree its occurrence depends on prior radiotherapy. Incidences of 0.80–9.75% have been reported based on retrospective data, but these studies generally lack a clear definition of PsP during follow-up [18,19]. Furthermore, the reported incidences in these studies cannot be reliably compared as it is unclear whether they are calculated only from lesions showing progression on imaging, or from all ICI-treated brain metastases.

PsP after ICI therapy seems to occur within several weeks to months after starting treatment [12,13,14,15,16,18,19]. This, however, has not been fully explored. The iRANO criteria suggest that any evidence of intracranial progression up to six months after starting ICI therapy requires an imaging follow-up of at least three months to exclude PsP [17]. The working group also states, however, that six months is not a hard threshold, highlighting the need for studies to validate this cut-off.

In this study, data is presented from a retrospective cohort of NSCLC and melanoma patients with brain metastases. Patients with signs of progression on MRI after starting ICI therapy were included, irrespective of prior radiotherapy. The focus was to establish the incidence of PsP specifically from lesions showing progressive changes on imaging. Inherent to the retrospective design of the study is that the reported incidence rate is necessarily an underestimation and can be seen as a ‘minimum’ valuation. This is because many lesions cannot be classified as PsP or TP retrospectively due to changes to treatment in response to signs of progression on imaging, as occurs frequently in day-to-day clinical practice. Of further interest in this study was the timing of PsP after starting ICI therapy and whether it is associated with various clinical variables. Greater awareness of these aspects will give clinicians a starting estimate for the scale of the clinical problem of PsP and inform radiologists of the likelihood of PsP when confronted with imaging changes after ICI therapy initiation.

## 2. Materials and Methods

### 2.1. Study Design and Participants

A single-institution retrospective cohort study, approved by the Medical Ethics Committee of the University Medical Center Groningen and registered centrally (9 June 2022; 2022/273; 202200340), was conducted. All screened patients were assessed for whether they had submitted a signed objection to the retrospective use of their data for research purposes. The electronic patient registry was screened using a pre-compiled database of melanoma patients with brain metastases diagnosed initially on either brain CT or MRI and seen as in- or out-patients between February 2011 and February 2022. Due to incompleteness of this database, missing patients were systematically screened with specified search terms using a data exploration tool within the electronic patient registry. Additionally, the electronic patient registry was screened using a pre-compiled database of all NSCLC patients treated with any ICI-containing treatment regimen between December 2014 and May 2022.

### 2.2. Inclusion and Exclusion Criteria

Patients with histopathologically proven primary or metastatic tumours were included if they showed progression of any intra-axial intracranial lesion that satisfied RANO-Brain Metastases (RANO-BM) criteria at any time point after the first infusion of ICI therapy. The analysis was focussed on individual lesions, so the RANO-BM criteria for progression were adapted so that they applied to a single lesion of interest as opposed to a set of target lesions [11]. For lesions that were already present prior to initiation of ICI therapy, progression was defined as an increase in longest diameter by at least 20.0% and 5 mm in the transverse or sagittal plane. New lesions after starting ICI therapy were also considered progressive and therefore suspect for either PsP or TP.

Patients were excluded in cases where there was no evidence of brain metastasis, no ICI therapy administration on record, only leptomeningeal or dural disease on MRI, an unclear ICI therapy start date, no evidence of intracranial progression since initiating ICI therapy, or insufficient radiological follow-up. Patients were also excluded (even if showing progression) in cases when the lesion was not measurable according to the RANO-BM criteria.

### 2.3. Collected Data

Collected data included patient age, patient sex, available molecular diagnostic information of the primary tumour or any metastatic site, type of ICI therapy, total number of infusions/cycles, duration of treatment, whether previous resection of the brain metastasis took place, the type and dosage of any previous brain radiotherapy (SRT or whole-brain radiotherapy (WBRT)), and the type and duration of other previous systemic treatment.

### 2.4. Nomenclature of Scan Time-Points and Lesions

The first scan showing evidence of progression for an individual lesion was termed the *progression scan*, following the iRANO criteria [17]. Following RANO-BM, the MRI closest to the start of ICI therapy was designated the *baseline scan*. The absolute and percentage changes in longest lesion diameter used to determine progression were calculated relative to the smallest diameter measurement on any scan after or immediately prior to ICI therapy [11,17]. This scan was termed the *nadir scan*.

Only lesions measurable according to RANO-BM criteria at the progression scan were included [11]. This was to allow for a reliable assessment of subsequent lesion response at the point of radiological follow-up (see ‘definitive diagnosis of included lesions’). Lesions were classified as ‘de novo’ or ‘pre-existent’. Pre-existent lesions were those already present regardless of tumour diameter prior to the start of ICI therapy. De novo lesions were newly emerged lesions after the start of ICI therapy. Furthermore, also registered was whether progression of lesions (RANO-BM criteria) was already observed on a pre-treatment scan if this was also just prior to the progression scan. This was to account for a trend in progression that may have started before ICI therapy.

### 2.5. Definitive Diagnosis of Included Lesions

The definitive diagnosis of PsP or TP after the progression scan was made according to either histopathological assessment after resection/biopsy or radiological follow-up. For lesions diagnosed by histopathology, the judgement of the board-certified pathologist at the time of resection or biopsy was deferred to. Following iRANO criteria, diagnosis by radiological follow-up was performed after three months, with the first scan at or immediately after this period termed the *definitive diagnosis scan* [17].

Lesions could be classified into one of three diagnostic groups at the definitive diagnosis scan [11]. (1) TP was assigned if the lesion showed further progression (additional increase of at least 20.0% and 5 mm in the transverse or sagittal plane) at the definitive diagnosis scan relative to the progression scan. (2) Non-classified (NC) was applied if a 20.0% and 5 mm diameter increase occurred after a change in or start of new ICI therapy or radiotherapy in the follow-up period. In this case, the possibility of recurrent PsP due to the new treatment interfered with a distinct diagnosis. Lesions were also designated NC if showing RANO-BM complete response (CR), partial response (PR), or stable disease (SD) together with a start of or change in radiotherapy, systemic therapy, or ICI therapy during follow-up. This interfered with the assessment of PsP due to possible anti-tumour effects at the definitive diagnosis scan. (3) PsP was assigned if a lesion showed RANO-BM CR, PR, or SD at the definitive diagnosis scan without initiation of new therapy or change in treatment after the progression scan (Figure 1).

### 2.6. Clinical Variables at the Progression and Definitive Diagnosis Scans

New-onset neurological symptoms relative to the nadir scan were recorded at the progression scan. The use and dose of dexamethasone because of neurological symptoms at the progression scan were also recorded. Dexamethasone prophylaxis administered before/around radiotherapy or systemic corticosteroids administered for other reasons, like chemotherapy, were not recorded. Similarly, further neurological deterioration and dexamethasone use and dose at the definitive diagnosis scan relative to the progression scan were documented.

To avoid bias, in cases where the same patient had two or more lesions with the same progression scan dates, only one lesion was included in this analysis. Where the same patient had two lesions with separate TP and PsP diagnoses and the same progression scan dates, only the TP lesion was included. Similarly, TP lesions were prioritised for inclusion over NC lesions for the same patient where the progression scan dates were the same.

### 2.7. Target, Non-Target, and New Lesions

In parallel with evaluating lesions individually, the overall intracranial disease burden was monitored using the RANO-BM criteria. A maximum of 5 measurable lesions at the baseline scan were registered as target lesions [11]. All other lesions were considered non-target lesions. First, diameters of all target lesions were measured and summed at all time points. The sum of diameters was then compared between scans to assess the response of the target lesions. Diameters of all non-target lesions were also tracked to assert whether they were unequivocally progressive or not at the progression and definitive diagnosis scans. Unequivocal non-target progression was only considered if all non-target lesions satisfied RANO-BM progression criteria at the relevant scans [11]. Lastly, the appearance of new lesions (according to the RANO-BM) and the disappearance of pre-existent lesions, both target and non-target, were monitored at the progression and definitive diagnosis scans.

### 2.8. MRI Protocols and Imaging Analysis

For tracking lesions from the baseline scan through to the definitive diagnosis scan, preferentially used were post-contrast T1-weighted 3D magnetisation prepared–rapid gradient echo (MPRAGE) sequences, which were available for 229/233 included lesions (98.3%) at the progression scan. The other four sequences used at the progression scan were a post-contrast T1-weighted 3D sampling perfection with application-optimised contrasts using a different flip-angle evolution (SPACE) sequence (n = 1), a post-contrast T1-weighted spin echo (SE) sequence (n = 1), and a post-contrast T1-weighted 3D turbo field echo (TFE) sequence (n = 2).

Scans were acquired on 3.0T Achieva (Philips, Eindhoven, the Netherlands) (n = 1), 1.5T Magnetom Aera (Siemens Healthineers, Erlangen, Germany) (n = 159), 1.5T Magnetom Avanto (Siemens) (n = 14), 1.5T Magnetom AvantoFit (Siemens) (n = 51), 1.5T Magnetom Espree (Siemens) (n = 1), 1.5T Magnetom Essenza (Siemens) (n = 1), 1.5T Ingenia (Philips) (n = 1), 3.0T Magnetom Prisma (Siemens) (n = 2), and 3.0T Magnetom Skyra (Siemens) (n = 3) scanners.

Image acquisition parameters were as follows: voxel size 0.16–1.10 × 0.16–1.10 × 0.16–1.10 mm, slice thickness 0.9–5 mm (5 mm: n = 1, 3 mm: n = 1, 1.1 mm: n = 1, 1 mm: n = 224, 0.9 mm: n = 6), repetition time (TR) 7.75–2300.00 ms, echo time (TE) 2.32–11.00 ms, inversion time (TI) 850–920 ms, flip angle 8–120°, and field-of-view (FoV) 201-230-256-256.

For each lesion, the longest diameter and longest perpendicular diameter (both in mm) were manually measured at relevant follow-up time points on both transverse and sagittal planes. For this, the measure and annotate function in SECTRA UniView (link to product website: https://medical.sectra.com/product/sectr-a-uniview/ (accessed on 16 July 2025)) (SECTRA AB, Linköping, Sweden) was used.

### 2.9. Statistical Analysis

All analyses were performed on a per lesion basis unless otherwise stated. For continuous variables, the Mann–Whitney U test was used after Shapiro–Wilk assessment confirmed non-normality in at least one group subjected to comparison. Median values with the interquartile range (IQR) were used as descriptive statistics. For categorical variables, the Fisher’s exact test was used to compare associations after the creation of contingency tables. Odds ratios (OR) were also calculated. Percentages were used as descriptive statistics. The lesion-specific incidence of PsP was calculated as follows: (number of lesions showing PsP at the progression scan)/(total number of lesions included in the study) × 100. The patient-specific incidence was calculated as follows: (number of patients with a lesion showing PsP at the progression scan)/(total number of patients included in the study) × 100.

The log-rank test was used to compare survival distributions. When a patient had two or more lesions, the lesion with the earliest progression scan date was used to determine the survival period. When a patient had two lesions with separate TP and PsP diagnoses, only the TP lesion was used. Similarly, TP and PsP lesions were also respectively prioritised for use over NC lesions in a single patient. For patients still alive at the point of data collection, administrative censoring was applied (follow-up cut-off date was the last day of data collection).

A significance level of 0.05 was used with two-tailed *p*-values. Data curation prior to analysis was performed with R (version 4.4.1) within RStudio (version 2024.04.02). All statistical analyses were performed with GraphPad Prism (version 10.3.1) (GraphPad Software, LLC, San Diego, CA, USA).

## 3. Results

### 3.1. Patient and Lesion Characteristics

Overall, 813 NSCLC and 381 melanoma patients were screened, of which 98 patients with a total of 233 lesions were enrolled (Figure 2) (Appendix A). Baseline characteristics of patients with TP and PsP are summarised in Table 1. About half of the lesions classified as TP (n = 20, 48.8%) and three quarters of the lesions classified as PsP (n = 16, 72.7%) were found in male patients. The median age was not significantly different between the TP (60.8 years, IQR = 51.9–65.4 years) and PsP groups (52.7 years, IQR = 50.4–68.7 years) (*p* = 0.44). For all lesions, the median interval between the baseline scan and start of ICI-therapy was 0.93 months (IQR 0.41–1.91 months).

### 3.2. Incidence of PsP Versus TP

Identified were 41 TP, 170 NC, and 22 PsP lesions; 3 PsP and 15 TP lesions were classified by histopathology, the others by follow-up imaging. Based on 233 lesions analysed, the incidence of PsP was 9.4%. The 22 PsP lesions were found in 17 patients, resulting in a patient-specific incidence of 17.3%.

Of the 22 PsP lesions, 10 lesions (45.5%) were irradiated prior to the progression scan, mostly with SRT (n = 9). In the TP group, 13 lesions (31.7%) received prior radiotherapy, again mostly with SRT (n = 10).

### 3.3. Timing of PsP and TP

There was no significant difference in the median interval between the start of ICI therapy and the progression scan between the TP (7.4 months, IQR = 2.5–13.3 months) and PsP groups (2.7 months, IQR = 1.9–9.8 months) (*p* = 0.07) (Figure 3A). The maximum interval observed in the PsP group was 34.1 months. Similarly, there was no significant difference in the median interval between administered radiotherapy and the progression scan between the TP (9.3 months, IQR = 7.6–16.3 months) and PsP groups (10.2 months, IQR = 3.3–15.5) (*p* = 0.70) (Figure 3B). The median interval between the start of ICI therapy and the progression scan was significantly higher in the TP group (6.4 months, IQR = 2.3–13.1 months) compared to PsP (2.5 months, IQR = 1.2–2.8 months) when analysing lesions without prior radiotherapy (*p* = 0.04) (Figure 3C). The difference between TP (10.9 months, IQR = 5.3–15.7 months) and PsP (9.1 months, IQR = 2.1–16.8 months) was not significant when only analysing lesions with prior radiotherapy (*p* = 0.52) (Figure 3C).

### 3.4. Clinical Differences Between PsP and TP

The proportion of patients treated with dexamethasone for neurological symptoms was significantly greater in the TP group at both the progression scan (51.4% TP versus 12.5% PsP) (*p* = 0.01, OR = 7.4) and the definitive diagnosis scan (62.2% TP versus 18.8% PsP) (*p* < 0.01, OR = 7.1) (Figure 4A,D). There was no significant difference in either the dexamethasone dose or the proportion of patients with neurological deterioration at both time points between the groups (Figure 4B,C,E,F).

The proportion of pre-existent to de novo lesions was not significantly different between the TP (58.5% pre-existent) and PsP groups (72.7% pre-existent) (*p* = 0.29, OR = 0.5). Differences in all other characteristics compared at both scan time points were not significantly different apart from the longest lesion diameters at the moment of definitive diagnosis (Table 2). The proportion of cases where disappearance or unequivocal progression of non-target lesions occurred was comparable between the groups at both time points (Appendix A).

### 3.5. Summed Target Lesion Changes at the Progression and Definitive Diagnosis Scans

The overall intracranial disease burden was monitored by tracking the summed diameters of target lesions at the nadir scan, progression scan, and definitive diagnosis scan. Target lesions were summed in the case of each of the 233 lesions included in the study except de novo lesions (no nadir scan available) and lesions diagnosed by histopathology (no definitive diagnosis scan available). In 100 out of these 127 (78.7%) individual lesions at the progression scan, the sum of target lesions also satisfied RANO-BM progression criteria (Figure 5A). There were 24 (18.9%) and 3 (2.4%) individual lesions where the summed target lesions at the progression scan satisfied criteria for SD and PR, respectively (Figure 5A).

At the definitive diagnosis scan, 19 out of 19 (100.0%) analysed cases of individual lesion PsP corresponded to the sum of target lesions showing SD or PR (RANO-BM) (Figure 5B). Five out of twenty-six (19.2%) cases of individual lesion TP corresponded to summed target lesions showing SD at the definitive diagnosis scan (RANO-BM) (Figure 5B). Similarly, target lesions were summed in the case of each of the 233 lesions included in the study except lesions diagnosed by histopathology (no definitive diagnosis scan available), explaining the smaller sample sizes for PsP (n = 19) and TP (n = 26).

### 3.6. Survival Differences Between PsP and TP

There was no significant overall survival difference between the PsP and TP groups, although the PsP group did trend towards improved overall survival (median overall survival 22.48 and 21.07 months for PsP and TP respectively) (*p* = 0.45) (Appendix A).

## 4. Discussion

This study gives insight into the incidence of PsP after ICI therapy at the point of initial suspicion of progression in melanoma and NSCLC patients with brain metastasis. Uniquely, the PsP incidence in lesions both with and without prior radiotherapy was analysed. Also examined was how PsP and TP lesions varied in terms of clinical presentation at various time points. PsP was found in 22 (17 patients) out of 233 lesions (98 patients), defining a lesion-specific incidence of 9.4% and a patient-specific incidence of 17.3%. Due to the large number of NC lesions, it follows that this incidence rate is necessarily an underestimation. It is more accurate therefore, as previously written, to see this as a ‘minimum incidence rate’. Of the 22 PsP lesions, 10 (45.5%) had prior radiotherapy. In the TP group, 13 (31.7%) had prior radiotherapy. This suggests that while radiotherapy may indeed increase the incidence of PsP after ICI therapy, it is not required for its onset. It is not clear from the results that the combination of radiotherapy and ICI therapy gives a higher incidence rate relative to ICI therapy only, as near equal lesion numbers were found in both groups. Nevertheless, this is the first study to specifically compare PsP incidence rates between patients treated with ICI therapy only and the combination of radiotherapy and ICI therapy.

Only few prior studies, mainly case reports, have specifically addressed the clinical problem of PsP after ICI therapy in patients with brain metastases [12,13,14,15,16]. Urban et al. observed an incidence of 9.8% among 123 patients [18]. Here, the incidence was calculated from all patients with intracranial disease treated with ICI therapy and followed with serial MRI. The current study defined the incidence only using lesions with progressive changes on MRI after the start of ICI therapy. The lower value reported by Urban et al. reflects this methodological difference. It is difficult to compare the findings of this work with the current study, as their study population was more heterogeneous, including gliomas and primary central nervous system lymphomas in addition to brain metastases. Furthermore, their study was ambiguous in terms of how lesion response criteria were defined.

A second study that strived to quantify PsP incidence after ICI therapy was by Hendriks et al., in which a patient specific value of 0.8% was reported in NSCLC patients with brain metastases [19]. Similarly, the authors defined incidence in terms of all patients with brain metastases treated with ICI therapy. Specific details on whether their strategy to define PsP corresponded with iRANO/RANO-BM criteria were lacking. It appears that only lesions showing subsequent shrinkage after progression were considered for PsP, not those showing stability with no further increase in longest lesion diameter (as the iRANO criteria stipulate). This detail likely makes this incidence value an underestimation of the true incidence. Granted, this can only be definitively known with available histopathology for all patients, which is rarely available. While the approaches adopted by both Urban et al. and Hendriks et al. offer valuable insights, the method presented in the current study for defining incidence is different in that it was calculated only from lesions showing signs of suspected progression. This is the most sensitive point of follow-up for a brain metastasis patient, as any uncertainty here can complicate downstream clinical decision-making. Understanding both lesion- and patient-specific incidence at this time point can help to inform whether current treatment should be continued or, conversely, if a change in treatment is required.

In clinical practice, SRT and ICI therapy are often used in combination [20,21]. The impact on PsP rates of this combination is not well known. While PsP was mostly observed within three months after starting ICI therapy in the current study, it also occurred up until 11 months without prior radiotherapy and up to 34 months with prior radiotherapy. The latter case might potentially be confounded by a delayed radiotherapy effect [22,23]. Indeed, no case of PsP occurring after 11 months took place without prior radiotherapy (see Appendix A). Of the 12 PsP lesions without prior radiotherapy, 11 occurred within six months after starting ICI therapy. Although this single case is not sufficient to suggest that clinicians should be aware of PsP throughout a follow-up period longer than the iRANO recommended threshold of six months, it does encourage further validation with a larger study cohort.

Reports in the literature indicate that with combined SRT–ICI therapy, the incidence of PsP increases compared to SRT given as monotherapy. One study reported a 20% PsP rate with combined SRT–ICI therapy compared to 5% in SRT alone [24]. In a separate retrospective cohort study, Rauch et al. tracked volumetric changes of brain metastases treated concomitantly with SRT and ICI therapy, SRT and targeted therapy, and SRT and ICI therapy in addition to targeted therapy [25]. Compared to the non-ICI-treated lesions, ICI-treated lesions were more likely to show increases in volume at the earliest point of follow-up after SRT (2 months). The majority of these lesions then regressed to baseline measurements at the next point of follow-up (4 months post-SRT). This may reflect higher rates of PsP within the ICI-treated groups, but could also be due to different patient and tumour characteristics leading to a delayed treatment response. Another study by Rahman et al. revealed that patients treated concomitantly with ICI therapy and SRT are more likely to have intracranial progression within 60 days after starting ICI therapy relative to patients not treated concomitantly (>30 days between SRT and ICI therapy) [26]. Again, while increased PsP rates in the concomitant group could have been driving this difference, the authors suggest that this is unlikely given that early progression was associated with worse overall survival. The concomitantly treated group had a significantly greater median age, however, which was likely also contributing to this. There were no differences in brain metastasis number or tumour burden between the two groups. Indeed, the results presented in the current study on the timing of PsP after starting ICI therapy suggest that it generally occurs within three months after starting ICI therapy. Although the cohort of the present study is not entirely comparable with that of Rahman et al., their study shows that early increases in lesion size cannot always be used as a readout for PsP. This emphasises the importance of careful clinicoradiological follow-up in each case of progression after ICI therapy.

In addition to incidence, the various clinical features associated with PsP and whether these could be used for differentiation from TP were determined. The results presented here indicate that there is no significant difference in the proportion of patients experiencing new-onset neurological deterioration between the PsP and TP groups, both at the progression (observed in 31.3% versus 54.1%) and definitive diagnosis scans (observed in 18.8% versus 35.1%). These observations align with previous reports showing a relatively high prevalence of new-onset neurological symptoms at PsP diagnosis. Out of 12 patients showing PsP in the cohort of Urban et al., 11 showed new-onset abnormal findings on neurological examination [18]. Of the six patients discussed in the available case reports, five presented with neurological deterioration [12,13,14,15,16]. There may be a disparity in the severity of these symptoms between patients with TP and PsP lesions. The proportion of patients requiring alleviation with dexamethasone was significantly higher for those with TP lesions at both time points in the present study’s cohort. This could not be explained by differences in the dimensions of lesions, as there was no difference in lesion longest diameter between TP and PsP at any follow-up point except for the definitive diagnosis scan. This is as expected given this study’s methodology, as TP lesions by definition showed further RANO-BM progression at the definitive diagnosis scan relative to the progression scan. Additionally, both the number of ICI therapy infusions and duration of treatment were not associated with development of PsP. Lastly, there was no significant difference in overall survival between patients with PsP and TP lesions. There was, however, a trend towards improved overall survival in the PsP group, and it is possible that this subanalysis lacked statistical power to carry this to a significant difference. Interestingly, however, similar findings of no difference in overall survival were also seen in a subanalysis by Umemura et al., a study primarily intended to evaluate the diagnostic accuracy of dynamic contrast-enhanced MRI for distinguishing PsP and TP after ICI therapy [27].

This study also has limitations. First, as stated previously, due to its retrospective nature, the initiation of or change in therapy between the progression and definitive diagnosis scans could not be controlled for. This implies that the calculated PsP incidence is likely an underestimation. The majority of the NC lesions were likely to have been cases of TP, but tumour effects of newly started treatment made it impossible to classify lesions definitively at follow-up (170 lesions could not be classified). This reflects clinical practice, where it is often not possible to deny a patient a change in therapy when confronted with progressive imaging changes despite theoretical uncertainty about whether this represents true tumour growth. Ultimately the only way to come to a definitive incidence rate is through prospective studies. These may be difficult to conduct from an ethical perspective (specifically regarding the justification for withholding therapy changes or performing a biopsy after progression on imaging to a patient in a research setting) without well-constructed retrospective studies as a foundation to suggest that the incidence of PsP can rationalise this clinical decision. The second limitation of this study is the unavoidable heterogeneity in the cohort in terms of prior systemic and radiotherapy treatment. The NSCLC cohort, for example, included patients treated with ICI therapy and chemotherapy in combination. It cannot be excluded that there are possible confounding effects of these different modalities on the incidence of PsP and that the incidence rates reported may not be replicated if these variables were more stringently controlled for.. Thirdly, it is recognised that it is not possible using radiological follow-up to distinguish between PsP and true tumour growth if the latter is followed by a delayed response to ICI therapy. This challenge is also recognised by the authors of the iRANO criteria [17]. Although these two entities are pathophysiologically distinct, they are very similar in terms of clinical behaviour. It therefore remains clinically applicable to group them under the heading of PsP, as applied here. Lastly, because of the fact that patients with measurable lesions at the progression scan were included specifically, smaller lesions (<10 mm longest diameter) are not represented in this study population. This may reduce the translatability of the results, as smaller brain metastases at baseline have been found to be more susceptible to PsP [28]. All of these limitations emphasise the need for prospective studies to validate and build on these findings. These may confirm the suspicion that the true incidence of PsP after ICI therapy in brain metastases is in fact greater than reported here.

## 5. Conclusions

This study reports a minimum 9.4% lesion-specific and 17.3% patient-specific incidence of PsP after ICI therapy in melanoma and NSCLC brain metastases. PsP without prior radiotherapy tends to occur within three to five months after starting ICI therapy, but this may extend to eleven months. The sole clinicoradiological variable that clearly distinguishes between patients with PsP and TP lesions at the point of progressive changes on MRI are that patients with TP are more likely to have neurological symptoms severe enough to require dexamethasone. These findings can give clinicians an indication as to the likelihood of PsP when confronted with progressive imaging changes after ICI therapy initiation. Considering that the results suggest this to be the minimum incidence rate, it is possible that this phenomenon is more widespread than is currently appreciated in neuro-oncology.

## Figures and Tables

**Figure 1 cancers-17-02425-f001:**
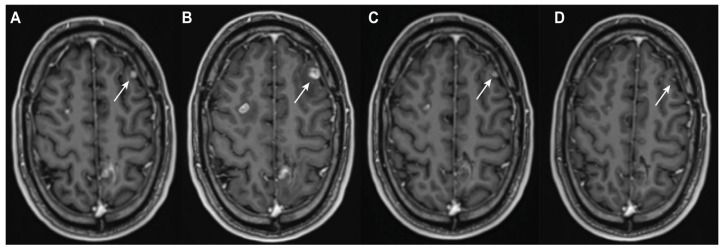
Case of pseudoprogression due to ICI therapy. Post-contrast T1-weighted imaging of a 38-year-old man with melanoma. The patient was treated for approximately four months with combination dabrafenib and trametinib after the diagnosis of diffuse cerebral metastatic disease. The left frontal lesion (white arrow) showed initial evidence of response ((**A**) nadir scan) and measured 6 × 5 mm in the transverse plane. ICI therapy (combination nivolumab–ipilimumab) was started. Diffuse progression was seen on MRI 7 days after the second dose, with the left frontal lesion increasing in size to 11 × 10 mm ((**B**) progression scan). A follow up scan 1 month later without any intervening treatment showed a partial response to 5 × 4 mm (**C**). The next follow-up scan, 4 months after the progression scan, showed a near complete response to 2 × 1 mm, again without any intervening treatment ((**D**) definitive diagnosis scan).

**Figure 2 cancers-17-02425-f002:**
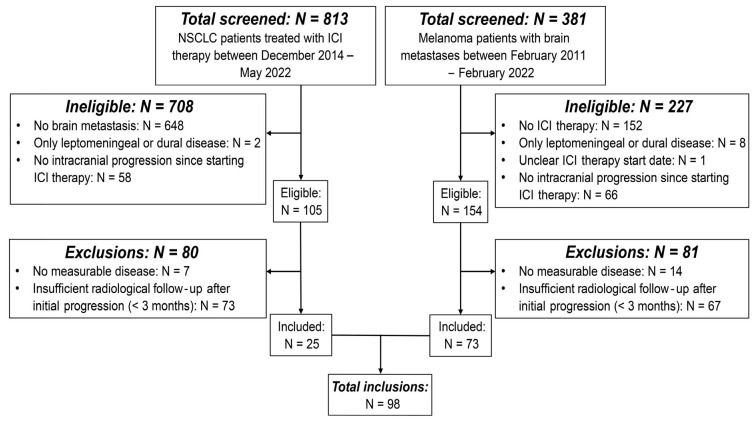
Flow chart of inclusions. Flow chart showing the patient screening and inclusion process. The flow chart outlines the reasons per patient group for initial exclusion due to ineligibility and subsequent exclusions. Abbreviations—NSCLC: non-small cell lung cancer; ICI: immune-checkpoint inhibitor.

**Figure 3 cancers-17-02425-f003:**
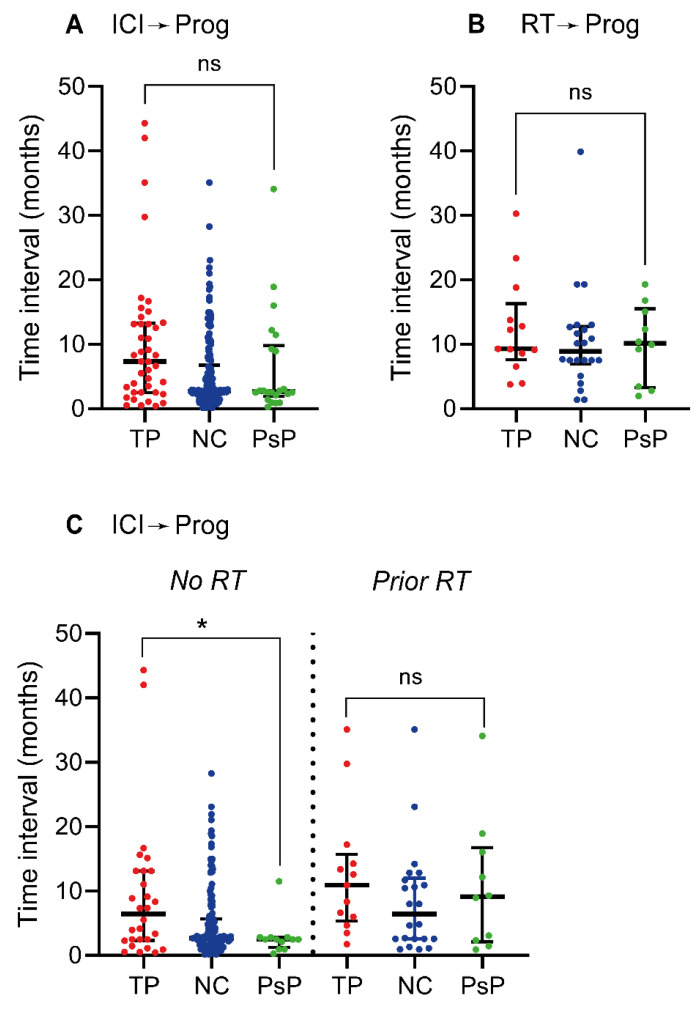
Timing of TP and PsP emergence after therapy. (**A**) Scatter plot of time interval in months between the start of ICI therapy and the point of initial progression—the progression scan. In the case of two separate ICI therapy treatment lines prior to the progression scan, the interval was calculated from the start of the second treatment line. (**B**) Time interval in months between the point of radiotherapy and the progression scan. For WBRT, the interval was calculated from the day of the last administered dose and for multiple separate treatments the interval was calculated from the last recorded treatment point. (**C**) Time interval in months between the start of ICI therapy and the progression scan for lesions without (left of vertical dotted line) and with (right of vertical dotted line) prior radiotherapy. The scatter plots have horizontal lines depicting the median and vertical error bars indicating the interquartile range; ns: non-significant (*p* ≥ 0.05), *: *p* < 0.05. Abbreviations—ICI: immune-checkpoint inhibitor; Prog: progression scan; RT: radiotherapy; TP: tumour progression; NC: non-classified; PsP: pseudoprogression.

**Figure 4 cancers-17-02425-f004:**
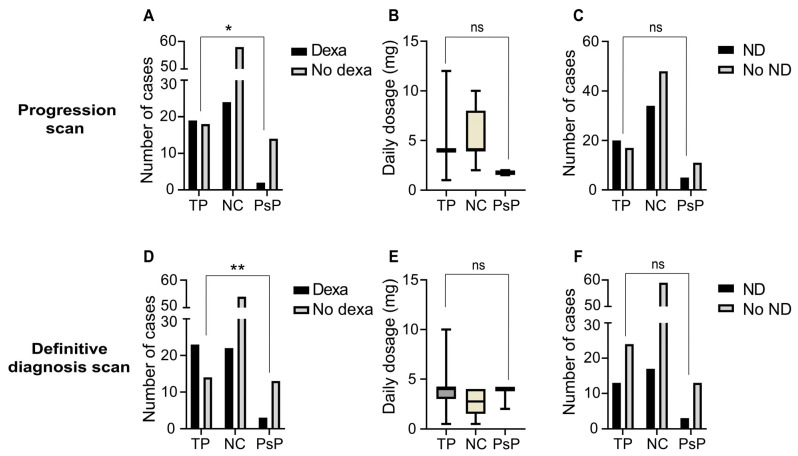
Clinical differences between TP and PsP groups at the progression and definitive diagnosis scans. (**A**) Bar plot showing the number of cases per diagnostic group that received dexamethasone for neurological symptoms at or around the point of initial progression—the progression scan. (**B**) Box-and-whisker plot indicating the daily dexamethasone dosage (mg) per case at or around the progression scan. (**C**) Bar plot showing the number of cases with neurological deterioration at or around the progression scan. (**D**–**F**) Bar plots and box-and-whisker plot showing data analogous to (**A**–**C**) but then at the definitive diagnosis scan. The horizontal bars show the median and the errors bars outline the interquartile range; ns: non-significant (*p* ≥ 0.05); *: *p* < 0.05; **: *p* < 0.01. Abbreviations—TP: tumour progression; NC: non-classified; PsP: pseudoprogression; Dexa: dexamethasone; ND: neurological deterioration.

**Figure 5 cancers-17-02425-f005:**
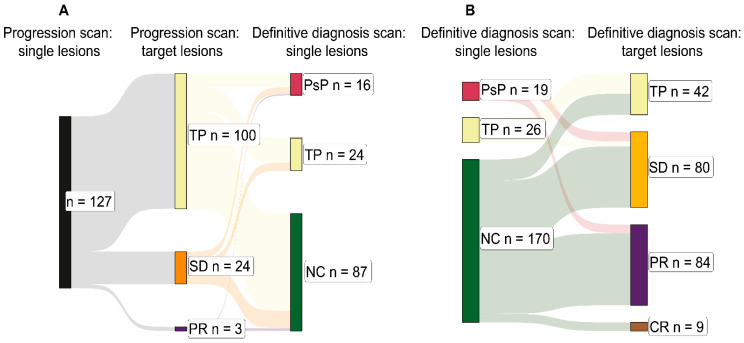
Total intracranial burden at initial progression and definitive diagnosis. (**A**) Sankey chart depicting the relationships between individual lesions under analysis at the point of initial progression—the progression scan (left: black)—and the suggested diagnosis according to the sum longest lesion diameter changes in all target lesions at the progression scan (middle: yellow, orange, and purple). This is followed by the definitive diagnosis for the individual lesions at the definitive diagnosis scan (right: crimson, yellow, and green). Target lesions were summed in the case of each of the 233 lesions included in the study except de novo lesions (no nadir scan available) and lesions diagnosed by histopathology (no definitive diagnosis scan available). This explains the smaller sample size of 127. (**B**) Sankey chart depicting the relationships between individual lesions under analysis at the definitive diagnosis scan (left: crimson, yellow, and green) and the suggested definitive diagnosis according to the sum longest lesion diameter changes in all target lesions at the definitive diagnosis scan (right: yellow, orange, purple, and brown). Similarly, target lesions were summed in the case of each of the 233 lesions included in the study except lesions diagnosed by histopathology (no definitive diagnosis scan available), explaining the smaller sample sizes for PsP (n = 19) and TP (n = 26). Abbreviations—TP: tumour progression; SD: stable disease; PR: partial response; PsP: pseudoprogression; NC: non-classified; CR: complete response.

**Table 1 cancers-17-02425-t001:** Basic lesion characteristics. Overview of lesion characteristics leading up to the progression scan. Lesions are presented according to definitive diagnosis as TP, NC, and PsP. Abbreviations—TP: tumour progression; NC: non-classified; PsP: pseudoprogression; IQR: interquartile range; NSCLC: non-small cell lung cancer; Gy: gray; SRT: stereotactic radiotherapy; WBRT: whole-brain radiotherapy; ^1^: two separate SRT treatments of 1 × 20 Gy at different times; ^2^: treated in external institution, from which there is a lack of data; ^3^: for detailed information on prior systemic treatment, see Appendix A.

	TP (N = 41)	NC (N = 170)	PsP (N = 22)	Mann–Whitney U *p-*Value TP–PsP
Age at progression scan (years), median (IQR)	60.8 (51.9–65.4)	59.1 (51.9–67.1)	52.7 (50.4–68.7)	0.44
Male sex, n (%)	20 (48.8)	109 (64.1)	16 (72.7)	-
Primary tumour type, n (%)				-
Melanoma	32 (78.0)	145 (85.3)	20 (90.9)	
NSCLC	9	25	2	
Adenocarcinoma	7	24	2	
Unknown subtype	2	1	0	
Prior resection, n (%)	1 (2.4)	3 (1.8)	0 (0.0)	-
Prior radiotherapy, n (%)	13 (31.7)	22 (12.9)	10 (45.5)	-
SRT	9 (22.0)	13 (7.6)	9 (40.9)	
WBRT	3 (7.3)	8 (4.7)	1 (4.5)	
SRT (2x) ^1^	0 (0.0)	1 (0.6)	0 (0.0)	
SRT + WBRT	1 (2.4)	0 (0.0)	0 (0.0)	
Total prior SRT dose administered, n (% of SRT courses)				-
1 × 20 Gy	8 (80.0)	12 (80.0)	9 (100.0)	
1 × 18 Gy	0 (0.0)	1 (6.7)	0 (0.0)	
3 × 8 Gy	2 (20.0)	1 (6.7)	0 (0.0)	
Unknown ^2^	0 (0.0)	1 (6.7)	0 (0.0)	
Total prior WBRT dose administered, n (% of WBRT courses)				
5 × 4 Gy	4 (100.0)	8 (100.0)	1 (100.0)	
Prior systemic or targeted therapy, n (%) ^3^				-
Yes	24 (58.5)	127 (74.7)	14 (63.6)	
Number of separate lines of ICI therapy given prior to progression scan, n (%)				-
1	33	146	21	
2	8 (19.5)	24 (14.1)	1 (4.5)	
Type of ICI therapy, n (%)				-
1st line of therapy given				
Ipilimumab	10 (24.3)	39 (22.9)	6 (27.3)	
Pembrolizumab	8 (19.5)	34 (20.0)	2 (9.1)	
Nivolumab	14 (34.1)	40 (23.5)	9 (40.9)	
Nivolumab/ipilimumab + nivolumab	0 (0.0)	1 (0.6)	2 (9.1)	
Nivolumab/ipilimumab	6 (14.6)	51 (30.0)	3 (13.6)	
Durvulumab	1 (2.4)	4 (2.4)	0 (0.0)	
Atezolizumab	2 (4.9)	1 (0.6)	0 (0.0)	
2nd line of therapy given				
Ipilimumab	0 (0.0)	0 (0.0)	1 (4.5)	
Pembrolizumab	2 (4.9)	13 (7.6)	0 (0.0)	
Nivolumab	6 (14.6)	7 (4.1)	0 (0.0)	
Nivolumab/ipilimumab	0 (0.0)	4 (2.4)	0 (0.0)	
Number of ICI infusions prior to progression scan, median (IQR)	5 (3–13)	3 (2–5)	4 (2–6)	0.34
ICI therapy duration prior to progression scan (months), median (IQR)	3.50 (1.39–7.04)	2.07 (0.70–2.93)	2.07 (0.81–2.94)	0.25
ICI therapy duration total (months), median (IQR)	3.80 (2.05–8.60)	2.07 (0.70–4.28)	3.42 (1.76–9.13)	0.83

**Table 2 cancers-17-02425-t002:** Further clinical comparisons at the progression and definitive diagnosis scans. Overview of additional clinical parameters at the points of initial progression and the definitive diagnosis. Lesions are presented according to definitive diagnosis as TP, NC, and PsP. Abbreviations—TP: tumour progression; NC: non-classified; PsP: pseudoprogression; IQR: interquartile range. ^1^: Analyses performed by summing the total cases where new lesions emerged at the progression/definitive diagnosis scans, prior to these scans and both at and prior to these. This total was then compared to the number of cases where no lesions emerged throughout the indicated follow-up points.

	TP (N = 41)	NC (N = 170)	PsP (N = 22)	Mann–Whitney U/Fisher’s Exact *p*-Value TP-PsP
Lesion type, n (%)				0.29
De novo	17 (41.5)	80 (47.1)	6 (27.3)	
Pre-existing	24 (58.5)	90 (52.9)	16 (72.7)	
Emergence of new lesions, n (%)				0.19 ^1^
At progression scan	22 (53.7)	109 (64.1)	8 (36.4)	
Prior to progression scan	1 (2.4)	11 (6.5)	2 (9.1)	
Both prior to and at progression scan	3 (7.3)	6 (3.5)	0 (0.0)	
Emergence of new lesions, n (%)				0.53 ^1^
At definitive diagnosis scan	7 (17.1)	12 (7.1)	4 (18.2)	
Prior to definitive diagnosis scan	0 (0.0)	21 (12.4)	1 (4.5)	
Both prior to and at definitive diagnosis scan	1 (2.4)	2 (1.2)	1 (4.5)	
Pattern of progression prior to progression scan and starting ICI therapy, n (%)				
Yes	6 (14.6)	14 (8.2)	3 (13.6)	>0.99 (<1.00)
Longest transverse diameter (mm), median (IQR)				
Baseline	7 (5–14)	5 (3–8)	7 (6–11)	0.80
Nadir	6 (5–13)	5 (3–7)	7 (4–11)	0.88
Progression	19 (12–32)	13 (11–18)	15 (11–18)	0.06
Definitive diagnosis	23 (19–30)	10 (5–16)	9 (4–11)	<0.01
Longest sagittal diameter (mm), median (IQR)				
Baseline	8 (5–16)	5 (3–8)	8 (5–10)	0.70
Nadir	6 (4–14)	5 (3–7)	7 (4–10)	0.82
Progression	18 (12–31)	14 (11–19)	15 (11–19)	0.13
Definitive diagnosis	23 (19–29)	10 (5–16)	9 (4–11)	<0.01
Longest sum transverse diameter target lesions (mm), median (IQR)				
Baseline	18 (8–34)	19 (6–31)	26 (7–50)	0.82
Nadir	16 (8–31)	15 (5–30)	24 (7–39)	0.93
Progression	31 (19–48)	38 (17–75)	32 (17–60)	0.79
Definitive diagnosis	34 (22–66)	24 (10–40)	12 (7–21)	<0.01
Longest sum sagittal diameter target lesions (mm), median (IQR)				
Baseline	19 (8–31)	19 (6–34)	29 (7–49)	0.69
Nadir	16 (8–30)	15 (6–34)	25 (7–42)	0.71
Progression	31 (18–54)	37 (16–75)	33 (16–55)	0.82
Definitive diagnosis	35 (22–68)	23 (11–41)	12 (7–23)	<0.01

## Data Availability

The collected raw data has been provided as a file in Appendix A.

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
