# Peer review of "Incidence and Clinical Features of Pseudoprogression in Brain Metastases After Immune-Checkpoint Inhibitor Therapy: A Retrospective Study"

_cancers, 2025, doi:10.3390/cancers17152425_

Round 1
Reviewer 1 Report (Previous Reviewer 1)
Comments and Suggestions for Authors
1. Reviewer Report
I thank the editors for the opportunity to review the article by Govaerts et al.
The article is well thought out and well designed. The introduction is short, which introduces the reader to the problems and issues related to brain metastases and pseudoprogression. The project has the approval of Medical Ethics Committee.
The selection of research methodology and research group is appropriate.
The presented results and discussion also do not raise any objections.
As a reviewer, I would like to draw your attention to a few issues:
I suggest changing the personal forms: we, our, to the impersonal form.
Line 178, what does the notation “iii” mean?
I think it would be a good idea to change the font in the figures to something more legible.
In my opinion, the manuscript (after minor revision) will be ready for publication.
Author Response
Reviewer #1
Comment 1: ‘I suggest changing the personal forms: we, our, to the impersonal form.’
Response:
We thank the reviewer for this suggestion. This has been changed throughout the manuscript.
Comment 2: ‘Line 178, what does the notation ‘iii’ mean?
Response:
This has been changed to number notation as 1, 2 and 3 for ease of comprehension (under ‘definitive diagnosis of included lesions’, lines 319-329 in the track-and-change manuscript version).
Comment 3: ‘I think it would be a good idea to change the font in the figures to something more legible’
Response:
We thank the reviewer for this suggestion, although it would be helpful if it was mentioned which figure and where specifically the difficulty arises. Throughout the manuscript and figures Arial is used, which is a widely known and standardised font recommended by many scientific journals (see for example under the Cancers ‘instructions for authors’ that the recommended font for graphical abstracts is Arial).
2. Reviewer Report
The authors have addressed and corrected all comments from the previous version. Congratulations.
Reviewer 2 Report (Previous Reviewer 2)
Comments and Suggestions for Authors
1. Reviewer Report
This study's primary conclusion—the incidence of PsP—suffers from low reliability because a vast number of cases were effectively excluded from the analysis as "non-classified." Furthermore, other conclusions drawn from small sample sizes and design limitations require cautious interpretation. Without addressing or acknowledging these issues, applying the findings of this study to clinical practice carries significant risks.
1. The Most Critical Issue : The existence of a vast "Non-Classified (NC)" Group. Of the 233 lesions analyzed, a staggering 73% (170 lesions) are designated as "Non-Classified (NC)." This group consists of cases where a definitive diagnosis of Pseudoprogression (PsP) or Tumor Progression (TP) could not be made because the treatment plan was changed after progression was suspected.
Severe selection bias: In a clinical setting, patients with severe symptoms or significant radiographic progression are more urgently moved to a change or intensification of treatment. Therefore, it is highly probable that this NC group is disproportionately composed of cases that were true tumor progression (TP). Underestimation of Incidence: As a result of this bias, the calculated incidence rate of PsP (9.4% per lesion), which is based only on the cases that were not excluded from the analysis (i.e., those classified as PsP or TP), is extremely likely to be a significant underestimation of the true incidence rate. The reliability of the study's most critical conclusion—the incidence rate—is severely compromised by this single point.
2. The study concludes that there was no significant difference in overall survival (OS) between the PsP and TP groups (p=0.45).
Why This is an Issue: Generally, PsP is considered a manifestation of a favorable immune response and is expected to correlate with a better prognosis, making this result counter-intuitive. Given the small number of cases in the PsP group (N=22 lesions / 17 patients), it is possible that the study simply lacked sufficient statistical power and failed to detect a real difference, leading to a "no difference" conclusion. The Kaplan-Meier curve itself shows a favorable trend for the PsP group, and this interpretation is insufficient. The study suggests a difference in the incidence of PsP between melanoma (10.2%) and non-small-cell lung cancer (NSCLC, 5.6%).
Why This is an Issue: The analysis for NSCLC is based on only 36 lesions. Drawing a conclusion about a difference in incidence rates between the two cancer types from this small sample size is statistically unreliable and premature. Based on an observation of PsP occurring 11 months after ICI initiation in a single case without prior radiotherapy, the study suggests that the iRANO criteria (6 months) might be too conservative. This assertion is based on the observation of just a single case. Deriving a general conclusion from a single exceptional case is a logical leap and constitutes very weak scientific evidence.
3. As this is a "retrospective cohort study," it inherently has the following limitations:
Inability to Control for Confounding Factors: Treatment methods (e.g., combination with chemotherapy in NSCLC) and patient backgrounds are heterogeneous, and factors that could influence the results (confounders) could not be eliminated.
Missing or Biased Information: A great deal of information, such as the decision-making process for treatment, could not be collected, which contributed to the large NC group.
Due to these limitations, the internal validity (the reliability of causal relationships) of the results is weakened.
4. Other, Minor Issues
Error in Table: The p-value listed in Table 2 as "p > 1.00" is an impossible value, which casts doubt on the data's reliability.
Inconsistent/Unclear Terminology: The use of non-standard expressions like "Pre-existent-now-measurable" may hinder reader comprehension.
Author Response
Reviewer #2
Comment 1: ‘The Most Critical Issue : The existence of a vast "Non-Classified (NC)" Group. Of the 233 lesions analyzed, a staggering 73% (170 lesions) are designated as "Non-Classified (NC)." This group consists of cases where a definitive diagnosis of Pseudoprogression (PsP) or Tumor Progression (TP) could not be made because the treatment plan was changed after progression was suspected.
Severe selection bias: In a clinical setting, patients with severe symptoms or significant radiographic progression are more urgently moved to a change or intensification of treatment. Therefore, it is highly probable that this NC group is disproportionately composed of cases that were true tumor progression (TP). Underestimation of Incidence: As a result of this bias, the calculated incidence rate of PsP (9.4% per lesion), which is based only on the cases that were not excluded from the analysis (i.e., those classified as PsP or TP), is extremely likely to be a significant underestimation of the true incidence rate. The reliability of the study's most critical conclusion—the incidence rate—is severely compromised by this single point.’
Response:
We are grateful for the reviewer’s critical feedback and close reading of the manuscript and its results. This is indeed an issue that we had recognised and mentioned as a limitation in the discussion (lines 636-641 in the original manuscript). We agree with the reviewer that this is a critical point and that it certainly deserves more attention in the manuscript. We have therefore made several additions under multiple subheadings throughout the study to emphasise the point that the incidence is likely underestimated because of the necessary exclusion of a large majority of the lesions. As the current incidence is unknown and the phenomenon of PsP in this context poorly understood, the here reported minimal incidence of 9.4% is still very informative to the field. The areas where this has additionally been specified are as follows:
- Abstract (lines 119-122 in the track-and-change manuscript version): ‘due to the large number of lesions that could not be classified, as is the case in clinical practice, the reported incidence in this study is likely an underestimation and can be seen as a ‘minimum’ incidence rate.’
- Introduction (lines 229-233 in the track-and-change manuscript version): ‘Inherent to the retrospective design of the study is that the reported incidence rate is necessarily an underestimation and can be seen as a ‘minimum’ valuation. This is because many lesions cannot be classified as PsP or TP retrospectively due to changes to treatment in response to signs of progression on imaging, as occurs frequently in day-to-day clinical practice.’
- Discussion (lines 531-534 in the track-and-change manuscript version): due to the large number of NC-lesions, it follows that this incidence rate is necessarily an underestimation. It is more accurate therefore, as previously written, to see this as a ‘minimum incidence rate’
Further, the original description of this limitation in the discussion has been altered and expanded on to further emphasise the point. This now reads as follows (lines 646-657 in the track-and-change manuscript version):
‘This study also has limitations. First, as stated previously, due to its retrospective nature the initiation of or change in therapy between the progression and definitive diagnosis scans could not be controlled for. This implies that the calculated PsP incidence is likely an underestimation. The majority of the NC lesions were likely to have been cases of TP, but tumour effects of newly started treatment made it impossible to classify lesions definitively at follow-up (170 lesions could not be classified). This reflects clinical practice, where it is often not possible to deny a patient a change in therapy when confronted with progressive imaging changes despite theoretical uncertainty about whether this represents true tumour growth.’
Additionally, throughout the manuscript where conclusions are stated, the wording has been modified to either read ‘minimum incidence rate’ (as in the conclusion, line 687) or ‘the incidence is at least...’ (as in the simple summary, line 75).
Unfortunately this limitation is not addressed by increasing the sample size of the study cohort, as the ratio of PsP and TP to NC-lesions would remain approximately the same. It is therefore a limitation inherent to the nature of the study and something that cannot be controlled for. Even in prospective studies this limitation would remain as the clinical decision to treat a patient would not be able be postponed to learn the true nature of the occurrence nor would a biopsy solely for research purposes to understand PsP incidence be ethically acceptable. As has now been added to the abstract and discussion (see above), this is in fact also similar to circumstances in daily clinical practice. Due to lack of data regarding this topic in brain metastases paired with imperfect imaging techniques for distinguishing between PsP and TP, clinicians are currently not be able to withhold intensification of or change in anti-tumour therapy when confronted with suspect imaging changes. This is despite the fact that theoretically such imaging changes may not reflect true tumour growth. We therefore view the large cohort of NC lesions as reflective of the current understanding around this topic.
The made modifications hopefully will reduce the concern of the reviewer that the direct application of the results to clinical practice is ‘high risk’. This is also not necessarily the intent of the manuscript given how little is known about the topic of pseudoprogression after immune checkpoint inhibitor therapy in brain metastases. We maintain however despite these limitations that the results presented are still of interest and can be of use to clinicians and future researchers. We stress that this study is the first to systematically attempt to assess the question of pseudoprogression incidence after immune checkpoint inhibition at the moment of suspect imaging changes. Careful attention has been given to assure conformity to RANO-BM/iRANO criteria. This makes the results reproducible and potentially expandable with larger datasets in the future. Further, the incidence rate of 9.4% that is given, more accurately now referred throughout as a ‘minimum value’, is still a potentially useful and important finding given the uncertainty around this topic. It suggests that the incidence rate is higher than may currently be appreciated. Ultimately the only way to come to a definitive incidence rate is through prospective studies. As already alluded to, these may be challenging to conduct from an ethical perspective (it is difficult to justify withholding therapy changes to a patient or performing a biopsy to establish the true incidence in a research setting) without well-constructed retrospective studies as a foundation to suggest that the incidence of PsP can rationalise this clinical decision.
The last sentence of the above paragraph was also added to the discussion section (lines 657-662 in the track-and-change manuscript version).
Comment 2: ‘The study concludes that there was no significant difference in overall survival (OS) between the PsP and TP groups (p=0.45).
Why This is an Issue: Generally, PsP is considered a manifestation of a favorable immune response and is expected to correlate with a better prognosis, making this result counter-intuitive. Given the small number of cases in the PsP group (N=22 lesions / 17 patients), it is possible that the study simply lacked sufficient statistical power and failed to detect a real difference, leading to a "no difference" conclusion. The Kaplan-Meier curve itself shows a favorable trend for the PsP group, and this interpretation is insufficient. The study suggests a difference in the incidence of PsP between melanoma (10.2%) and non-small-cell lung cancer (NSCLC, 5.6%).
Why This is an Issue: The analysis for NSCLC is based on only 36 lesions. Drawing a conclusion about a difference in incidence rates between the two cancer types from this small sample size is statistically unreliable and premature. Based on an observation of PsP occurring 11 months after ICI initiation in a single case without prior radiotherapy, the study suggests that the iRANO criteria (6 months) might be too conservative. This assertion is based on the observation of just a single case. Deriving a general conclusion from a single exceptional case is a logical leap and constitutes very weak scientific evidence.’
Response:
Again, we thank the reviewer for this critical feedback. This is a challenging area because to our knowledge the difference in survival between patients with PsP and TP in brain metastases after immune checkpoint inhibitor therapy specifically has not been widely investigated (as cited in the manuscript, there is a study by Umemura et al (DOI: 10.1007/s11060-019-03379-6), where no significant difference was found). Nevertheless, we fully appreciate the rationale given by the reviewer and have decided in response to revert Figure 6 to a supplemental file as Supplemental Figure 3 so that it is highlighted less clearly as a primary result. Additionally, we modified the discussion to bring more attention to the fact that the lack of a survival difference could have been due to insufficient statistical power as follows (lines 639-642 in the track-and-change manuscript version):
‘Lastly, there was no significant difference in overall survival between patients with PsP- and TP lesions. There was however a trend towards overall survival in the PsP group, and it is possible that this sub analysis lacked statistical power carry this to a significant difference.’
We are fully in agreement with the reviewer with regards to the valid point about the difference in incidence rates between the two cancer types. The areas where this was written have been omitted throughout the manuscript and the incidence is now only presented as being from the cohort as a whole.
We are additionally in agreement with the reviewer with regards to the point made about the single case of PsP occurring at 11 months after ICI initiation. This passage in the discussion has now been reformulated as follows (lines 585-589 in the track-and-changed manuscript version):
‘Although this single case is not sufficient to suggest that clinicians should be aware of PsP throughout a follow-up period longer than the iRANO recommended threshold of six months, it does encourage further validation with a larger study cohort.’
We hope that with these adjustments the conclusions drawn from the data have been nuanced and placed in the appropriate context.
Comment 3: ‘As this is a "retrospective cohort study," it inherently has the following limitations:
Inability to Control for Confounding Factors: Treatment methods (e.g., combination with chemotherapy in NSCLC) and patient backgrounds are heterogeneous, and factors that could influence the results (confounders) could not be eliminated.
Missing or Biased Information: A great deal of information, such as the decision-making process for treatment, could not be collected, which contributed to the large NC group.
Due to these limitations, the internal validity (the reliability of causal relationships) of the results is weakened.’
Response:
Again, we appreciate the author taking the time to evaluate the work critically and provide useful feedback. This is a pertinent comment and a recognised weakness of the study. To a certain degree the point made by the reviewer is inherent to all retrospective studies. Controlling for prior chemotherapy, for example, would have reduced the sample size further to an unacceptable level and potentially exacerbated the effect described by the reviewer in comment 1 (if PsP lesions were excluded). This is a weakness that in the original version of the manuscript was fully addressed in the discussion as follows (line 641-646 in the original manuscript version):
‘The second limitation of this study is the unavoidable heterogeneity in the cohort in terms of prior systemic and radiotherapy treatment. The NSCLC cohort for example, included patients treated with ICI-therapy and chemotherapy in combination. We cannot exclude possible confounding effects of these different modalities on the incidence of PsP and that the incidence rates reported may not be replicated if these variables were more stringently controlled for.’
We would argue that the confounding/effect modification of chemotherapy in NSCLC or BRAF-inhibition in melanoma is limited to a degree by the fact that pseudoprogression in brain metastases after these treatment modalities (in the case of melanoma) has not to our knowledge been described on a study population level. Regardless we fully understand the reasoning of the reviewer and accept that potential effects of these factors on the study outcome cannot be excluded.
With regards to the comment on the fact that the decision-making process for treatment was missing, this is indeed something that was discussed during study conception. We made the decision to not include such information for two reasons:
- It was difficult to standardise on a population-level what the specific criteria would be for a decision to qualify as being classifying for TP. This was made even more difficult by the clinical records often being challenging to interpret retrospectively. Such decisions are often made based on subjective interpretation of the clinical and radiological data using the expert opinion of the treating physician. Standardisation was difficult as this treating physician changed from patient to patient.
- It is difficult to envisage a scenario where the decision to change or intensify treatment would lead to a lesion being classified retrospectively as PsP as opposed to TP. In other words, classifying lesions retrospectively based on the decision making as read in clinical reports would not lead to a reclassification of any of the NC lesions as PsP lesions. This would therefore not affect the primary conclusion of the study, the incidence of PsP. Granted, if specific criteria for classification based on the clinical reports were devised, the NC group would be reduced in size as more lesions could have been classified as TP. Nevertheless, the uncertainty for these lesions about the diagnosis would remain given the necessarily complicating effect of a newly introduced treatment. This, in combination with point 1, led us to choose a more ‘secure’ approach to attain a more definitive cohort of TP and PsP lesions despite the negative effect of this on sample size.
Comment 4: ‘Other, Minor Issues
Error in Table: The p-value listed in Table 2 as "p > 1.00" is an impossible value, which casts doubt on the data's reliability.
Inconsistent/Unclear Terminology: The use of non-standard expressions like "Pre-existent-now-measurable" may hinder reader comprehension.’
Response:
We thank the reviewer for also pointing out minor issues in addition to the more important points made above.
The p-value in Table 2 has now been corrected as ‘> 0.99 (< 1.00)’. This occurred due to premature rounding of the p-value from the statistics output in GraphPad.
We agree that the term ‘pre-existent-now-measurable’ is non-standard and could lead to confusion. We have therefore merged the ‘pre-existent’ and ‘pre-existent-now-measurable' category to a single group simply named ‘pre-existent’. This change has been implemented throughout the manuscript, including Table 2 (with descriptive statistics also duly modified).
2. Reviewer Report
I agree with this paper.
This manuscript is a resubmission of an earlier submission. The following is a list of the peer review reports and author responses from that submission.
Round 1
Reviewer 1 Report
Comments and Suggestions for Authors
I thank the editors for the opportunity to review the article by Govaerts et al.
The article is well thought out and well designed. The introduction is short, which introduces the reader to the problems and issues related to brain metastases and pseudoprogression. The project has the approval of Medical Ethics Committee.
The selection of research methodology and research group is appropriate.
The presented results and discussion also do not raise any objections.
As a reviewer, I would like to draw your attention to a few issues:
I suggest changing the personal forms: we, our, to the impersonal form.
Line 178, what does the notation “iii” mean?
I think it would be a good idea to change the font in the figures to something more legible.
In my opinion, the manuscript (after minor revision) will be ready for publication.
Reviewer 2 Report
Comments and Suggestions for Authors
This study's primary conclusion—the incidence of PsP—suffers from low reliability because a vast number of cases were effectively excluded from the analysis as "non-classified." Furthermore, other conclusions drawn from small sample sizes and design limitations require cautious interpretation. Without addressing or acknowledging these issues, applying the findings of this study to clinical practice carries significant risks.
1. The Most Critical Issue : The existence of a vast "Non-Classified (NC)" Group. Of the 233 lesions analyzed, a staggering 73% (170 lesions) are designated as "Non-Classified (NC)." This group consists of cases where a definitive diagnosis of Pseudoprogression (PsP) or Tumor Progression (TP) could not be made because the treatment plan was changed after progression was suspected.
Severe selection bias: In a clinical setting, patients with severe symptoms or significant radiographic progression are more urgently moved to a change or intensification of treatment. Therefore, it is highly probable that this NC group is disproportionately composed of cases that were true tumor progression (TP). Underestimation of Incidence: As a result of this bias, the calculated incidence rate of PsP (9.4% per lesion), which is based only on the cases that were not excluded from the analysis (i.e., those classified as PsP or TP), is extremely likely to be a significant underestimation of the true incidence rate. The reliability of the study's most critical conclusion—the incidence rate—is severely compromised by this single point.
2. The study concludes that there was no significant difference in overall survival (OS) between the PsP and TP groups (p=0.45).
Why This is an Issue: Generally, PsP is considered a manifestation of a favorable immune response and is expected to correlate with a better prognosis, making this result counter-intuitive. Given the small number of cases in the PsP group (N=22 lesions / 17 patients), it is possible that the study simply lacked sufficient statistical power and failed to detect a real difference, leading to a "no difference" conclusion. The Kaplan-Meier curve itself shows a favorable trend for the PsP group, and this interpretation is insufficient. The study suggests a difference in the incidence of PsP between melanoma (10.2%) and non-small-cell lung cancer (NSCLC, 5.6%).
Why This is an Issue: The analysis for NSCLC is based on only 36 lesions. Drawing a conclusion about a difference in incidence rates between the two cancer types from this small sample size is statistically unreliable and premature. Based on an observation of PsP occurring 11 months after ICI initiation in a single case without prior radiotherapy, the study suggests that the iRANO criteria (6 months) might be too conservative. This assertion is based on the observation of just a single case. Deriving a general conclusion from a single exceptional case is a logical leap and constitutes very weak scientific evidence.
3. As this is a "retrospective cohort study," it inherently has the following limitations:
Inability to Control for Confounding Factors: Treatment methods (e.g., combination with chemotherapy in NSCLC) and patient backgrounds are heterogeneous, and factors that could influence the results (confounders) could not be eliminated.
Missing or Biased Information: A great deal of information, such as the decision-making process for treatment, could not be collected, which contributed to the large NC group.
Due to these limitations, the internal validity (the reliability of causal relationships) of the results is weakened.
4. Other, Minor Issues
Error in Table: The p-value listed in Table 2 as "p > 1.00" is an impossible value, which casts doubt on the data's reliability.
Inconsistent/Unclear Terminology: The use of non-standard expressions like "Pre-existent-now-measurable" may hinder reader comprehension.